# The Combination of Vascular Endothelial Growth Factor A (VEGF-A) and Fibroblast Growth Factor 1 (FGF1) Modified mRNA Improves Wound Healing in Diabetic Mice: An Ex Vivo and In Vivo Investigation

**DOI:** 10.3390/cells13050414

**Published:** 2024-02-27

**Authors:** Sandra Tejedor, Maria Wågberg, Cláudia Correia, Karin Åvall, Mikko Hölttä, Leif Hultin, Michael Lerche, Nigel Davies, Nils Bergenhem, Arjan Snijder, Tom Marlow, Pierre Dönnes, Regina Fritsche-Danielson, Jane Synnergren, Karin Jennbacken, Kenny Hansson

**Affiliations:** 1Research and Early Development, Cardiovascular, Renal and Metabolism (CVRM), BioPharmaceuticals R&D, AstraZeneca, 431 50 Gothenburg, Swedenclaudia.correia@astrazeneca.com (C.C.); karin.avall@astrazeneca.com (K.Å.); karin.jennbacken@astrazeneca.com (K.J.); 2Systems Biology Research Center, School of Bioscience, University of Skövde, 541 28 Skövde, Sweden; pierre@scicross.com (P.D.); jane.synnergren@his.se (J.S.); 3Imaging and Data Analytics, Clinical and Pharmacological Safety Science, BioPharmaceuticals R&D, AstraZeneca, 431 50 Gothenburg, Sweden; leif.hultin@astrazeneca.com; 4Advanced Drug Delivery, Pharmaceutical Science, BioPharmaceuticals R&D, AstraZeneca, 431 50 Gothenburg, Sweden; michael.lerche@astrazeneca.com (M.L.); nigel.davies@astrazeneca.com (N.D.); 5Alliance Management, Business Development and Licensing, BioPharmaceuticals R&D, AstraZeneca, Waltham, MA 02451, USA; 6Discovery Sciences, BioPharmaceuticals R&D, AstraZeneca, 431 50 Gothenburg, Sweden; arjan.snijder@astrazeneca.com (A.S.);; 7SciCross AB, 541 35 Skövde, Sweden; 8Department of Molecular and Clinical Medicine, Institute of Medicine, The Sahlgrenska Academy, University of Gothenburg, 405 30 Gothenburg, Sweden

**Keywords:** diabetes, diabetic foot ulcer, angiogenesis, revascularization, VEGF-A, FGF1, wound healing

## Abstract

Background: Diabetic foot ulcers (DFU) pose a significant health risk in diabetic patients, with insufficient revascularization during wound healing being the primary cause. This study aimed to assess microvessel sprouting and wound healing capabilities using vascular endothelial growth factor (VEGF-A) and a modified fibroblast growth factor (FGF1). Methods: An ex vivo aortic ring rodent model and an in vivo wound healing model in diabetic mice were employed to evaluate the microvessel sprouting and wound healing capabilities of VEGF-A and a modified FGF1 both as monotherapies and in combination. Results: The combination of VEGF-A and FGF1 demonstrated increased vascular sprouting in the ex vivo mouse aortic ring model, and topical administration of a combination of VEGF-A and FGF1 mRNAs formulated in lipid nanoparticles (LNPs) in mouse skin wounds promoted faster wound closure and increased neovascularization seven days post-surgical wound creation. RNA-sequencing analysis of skin samples at day three post-wound creation revealed a strong transcriptional response of the wound healing process, with the combined treatment showing significant enrichment of genes linked to skin growth. Conclusion: f-LNPs encapsulating VEGF-A and FGF1 mRNAs present a promising approach to improving the scarring process in DFU.

## 1. Introduction

Diabetes is a disease that increases in prevalence worldwide, and despite progress in treatment, a substantial unmet medical need remains. Diabetes also causes the development of long-term complications like diabetic foot ulcers (DFU). Moreover, severe consequences of DFU include chronic infections, amputations, decreased ambulatory activity, and worsening of comorbidities that together lead to increased mortality. Large efforts to address DFU have been made, but effective treatments to improve the healing of this type of wound are still lacking [1,2].

One of the most important approaches to finding an effective treatment for DFU has been to target the causes of the malfunctioning wound healing process. Suboptimal revascularization during the wound healing process, leading to tissue ischemia, is one of these underlying causes [3]. Therefore, stimulation of angiogenesis has been suggested and investigated as a potential way to target impaired wound healing [4,5,6,7,8]. Vascular endothelial growth factor A (VEGF-A); most often in the isoform with 165 amino acids, VEGF-A165) is a key player in the revascularization process, improving wound area oxygenation. Previous preclinical models have observed an improved wound healing process after administration of recombinant proteins or mRNA-based therapies [9,10]. However, randomized controlled clinical trials have not shown the same positive outcomes to support efficacy in patients [11,12]. The reasons for the failed clinical trials with VEGF-A therapies are not fully understood, but possible factors include suboptimal local concentration [12].

To overcome the disadvantages following the use of gene therapies and recombinant proteins, we developed a chemically modified mRNA optimized to ensure efficient production of the VEGF-A 165 protein [13]. This chemically modified mRNA is non-immunogenic and allows efficient and transient expression of the target protein [14,15,16]. VEGF-A modified mRNA has preclinically been shown to improve wound healing in diabetic mice after intradermal injections and has been clinically investigated in patients with type-2 diabetes and in patients undergoing open heart surgery in the following clinical trials: NCT02935712 and NCT03370887 (ClinicalTrials.gov, access dates: 18 January 2020 and 6 August 2021, respectively) [13,17,18]. Although promising, the functional pharmacodynamic effects and the potential therapeutic benefits of this emerging therapeutic on the microvasculature in a diabetic wound environment need further characterization. Also, combination with other growth factors complementing the VEGF-A activity to achieve additive effects is a logical follow-up to single-factor treatment, as is the improvement of the delivery and duration of the exposure in the near vicinity of the wound. To address these two aspects, fibroblast growth factor 1 (FGF1) was identified as a potential combination candidate for VEGF-A due to its biological effects on wound healing experimental models, where an improvement in re-epithelialization, keratinocyte proliferation, extracellular matrix generation, and proangiogenic activity has been observed [14,16]. In fact, FGF1 has been previously investigated in patients with chronic diabetic wounds as a monotherapy (NCT00425178) [19]. Moreover, FGF1 has also demonstrated \ enhanced activity in wounds infected by bacteria compared with FGF2 [19,20,21,22]. Interestingly, previous published data showed that sequence modifications in FGF1 increase its in vivo half-life and secretion in physiological conditions [23,24,25,26]. Based on the above, we have used a combination of modified mRNA consisting of VEGF-A and a modified version of FGF1, which includes point mutations, to produce a molecule with enhanced stability and biological activity.

Previously, mRNA molecules have shown therapeutic potential in a range of applications. To overcome the challenges for these molecules to reach specific targets and escape from the endosomal system, an efficient and safe delivery formulation is needed [27]. Lipid nanoparticles (LNPs) are an efficient formulation for therapeutic mRNA delivery to the target tissue [27,28]. Previous studies have tested the local delivery of VEGF-A-modified mRNA to diabetic wounds in mice in vivo. However, those experiments have been performed with buffer-formulated mRNA, and preparations have been injected intradermally, which is challenging as an administration method and might lead to exposure of the mRNA to RNAse degradation. Consequently, the use of LNPs could solve the disadvantages identified in previous works. In fact, the potential of LNPs as drug vehicles has been backed up by the US Food and Drug Administration (FDA), with the mRNA-based severe acute respiratory syndrome coronavirus 2 (SARS-CoV-2) vaccine being the most known example [29,30]. Another example of an FDA-approved non-vaccine drug using LNPs is the first RNA interference therapeutic for hereditary transthyretin amyloidosis (hATTR) [31].

The purpose of this study was to investigate if the use of LNPs carrying VEGF-A or FGF-1 mRNAs alone or in combination could promote angiogenesis and skin regeneration in an ex vivo model of adult mouse aortic rings and an in vivo wound healing model in diabetic mice.

## 2. Materials and Methods

### 2.1. Generation of Recombinant Purified Human FGF1 Proteins

Three variants of human FGF1 protein were designed and synthesized: (1) FGF1 (1–155) WT, (2) FGF1 (1–155) MUT, and (3) FGF1 (22–155) MUT. The first protein corresponds to the wild-type protein. The second and third were designed with four-point mutations: Q40P, S47I, H93G, and C117S. In FGF1 (22–155) MUT, the sequence of the signal peptide of the protein was removed and substituted by a classical signal sequence used in previous studies [26]. A schematic view of the three proteins and their peptide sequence is shown in Figure 1A and Appendix A, respectively. Human FGF1 protein variants were expressed with an N-terminal hepta HN-tag followed by a TEV-protease cleavage site in *E. coli* BL21 Star from pET24 plasmids. Cells were grown in terrific broth supplemented with 0.1% lactose for 4 h at 37 °C, followed by 16 h at 18 °C. Cells were lysed using sonication in a 2× Phosphate buffered saline (PBS) solution with 10% (*v*/*v*) glycerol, 20 mM imidazole, 1 mg/mL lysozyme, protease inhibitors, and 2.5 U/mL benzonase. Cell debris and insoluble material were pelleted by centrifugation for 45 min at 22k rpm in a JA25.50 Beckman rotor. FGF1 was purified using an automated Phynexus purification system with 160 microliter IMAC resin tips (Biotage, Sweden). To remove impurities and endotoxins, after protein immobilization, the resin was washed extensively with a buffer containing 2×PBS, 20 mM imidazole, and 1% (*w*/*v*) Triton X-100, followed by washing with 2×PBS and 20 mM imidazole. Proteins were eluted with 2×PBS with 300 mM imidazole. The N-terminal tag was removed by TEV protease digestion, followed by size exclusion chromatography using BioRAD SEC 70 column equilibrated in a buffer of 2×PBS, 10% (*v*/*v*) glycerol, 5mM DTT, 10 mM (NH_4_)_2_SO_4_, and 0.1 mM EDTA. Low endotoxin levels were confirmed in all protein preparations using an EndoSafe PTS endotoxin testing system (Charles Rivers, Wilmington, MA, USA). Thermal stability of the various FGF1 variants was determined using SYPRO orange and Lightcyler 480 systems (Roche, Basel, Switzerland).

### 2.2. Ex Vivo Aortic Ring Assay

Animal work was performed in accordance with the National Institute of Health (NIH) guidelines for use of experimental animals, and the study protocol was approved by the Animal Ethics Committee in Gothenburg (Ethical application number EA 2017_001173). Aortic ring assay was performed following the method described in a paper by Baker M. et al. [32]. *C57BL/6* female adult mice were obtained from Charles River Laboratories (Wilmington, MA, USA). Thoracic aortae were first dissected from 20- to 22- week-old mice, and a set of forceps was used to remove the fatty layer. Then, each arteria was cut into 3–5 mm rings. Thereafter, they were incubated in serum starvation media (Opti-MEM + GlutaMAX-I media containing 1% penicillin/streptomycin mix (Gibco, New York, NY, USA) overnight to equilibrate their growth factor responses and promote a uniform baseline state. The next day, rings were embedded in reduced-growth factor Matrigel (Corning Inc., Corning, NY, USA) and fed in growth media (Opti-MEM + GlutaMAX-I media + 2.5% FBS + 1% penicillin/streptomycin mix) supplemented with the treatments, which were refreshed every two days. The effect on microvessel sprouting of VEGF-A (link to the peptide sequence: NP_001165097 (R&D Systems, Minneapolis, MN, USA), the three recombinant purified FGF1 variants produced as described above (FGF1 (1–155) WT, FGF1 (1–155) MUT, and FGF1 (22–155) MUT), and the combination of VEGF-A + FGF1 (22–155) MUT proteins were evaluated over one week, where the following protein concentrations were used: FGF1 variants (ng/mL: 25, 50, 100), VEGF-A (ng/mL: 2, 10, 50). Two combinations (C1 and C2) of VEGF-A and FGF1 (22–155), MUT were used: C1 (FGF1 25 ng/mL + VEGF 10 ng/mL) and C2 (FGF1 50 ng/mL + VEGF 10 ng/mL). Sprouting area was measured using ImageJ 2 software (NIH, Madison, WI, USA).

### 2.3. mRNA and LNP Formulation

VEGF-A and FGF1 mRNAs were synthesized in vitro by T7 RNA-polymerase-mediated transcription from a linearized DNA template, as described before by our group [13]. LNPs were formulated with MC3 (as ionizable lipid), distearoylphosphatidylcholine, cholesterol, and 1,2-dimyristoyl-sn-glycero-3-phosphoethanolamine-N-[methoxy(polyethyleneglycol)-2000] at a molar ratio of 50, 10, 38.5, and 1.5, respectively, and a total lipid:mRNA ratio of 10:1 and a nitrogen:phosphate (N:P) ratio of 3, as previously reported by our team [33]. Briefly, an ethanolic solution of the lipid components (total lipid concentration: 12.5 mM) and a solution of the mRNA in RNase-free citrate buffer (pH 3, 50 mM) were mixed at a ratio of 1:3, respectively, at a total flow rate of 12 mL/min using a NanoAssembler (Precision NanoSystems, Vancouver, BC, Canada). Following microfluidic mixing, the LNPs were dialyzed overnight at 4 °C against 1000× sample volume of PBS (pH 7.4) using dialysis cassettes with a molecular weight cut-off of 10,000 (Thermo Fisher Scientific, Waltham, MA, USA). The resulting formulations were characterized in terms of size and polydispersity by dynamic light scattering using a Zetasizer Nano ZS (Malvern Instruments, Worcestershire, UK), and the encapsulation and concentration of mRNA in the LNP formulations were determined using the Ribo-Green assay [34]. For the studies reported, LNPs typically had a particle size (Z average) of 70–90 nm, polydispersity index of <0.1, and encapsulation efficiency of ≥97%.

### 2.4. Wound Healing In Vivo Model

Animal work was performed in accordance with the National Institute of Health (NIH) guidelines for use of experimental animals, and the study protocol was approved by the Animal Ethics Committee in Gothenburg (Ethical application number EA 2017_000687). Nine- to ten-week-old male *B6.BKS(D)-Leprdb/J* (db/db mice) were purchased from Jackson Laboratory, US, and were acclimatized for at least 5 days before experiments started. Prior to wound healing model created by surgery, the mice were fasted for 4 h and glucose concentration in blood was measured using an Accu-Check Active Blood Glucose Glucometer Kit. Blood glucose was measured on blood drawn from tail vein, and the mice were randomized into experimental groups based on blood glucose values and body weight. A total of 13 animals per group were included in the study, and they were divided into 2 sets to separate animals. Samples from skin were taken at different time points from the animals where an imaging follow-up was performed, as previously conducted by our team [18]. Wound healing area and angiogenesis experiments were measured in 7 animals per group for 17 days, while protein expression levels at two different time points (3 and 7 days after wound healing induction) were measured in 6 animals per group, obtaining data from 3 animals for each time point. Mice were anesthetized with a 2% inhalable isoflurane (Piramal Critical Care, Bethlehem, PA, USA)/oxygen mixture, and the dorsal side of the mice was shaved, depilated, and cleaned with Viscutan and ethanol prior to the wound induction surgeries. A 1 centimeter-diameter, full-thickness skin wound (including dermis and epidermis) was surgically made on the dorsum of the mouse, and the wounds were covered with a Tegaderm dressing. An analgesic (buprenorphine 0.08 mg/kg) was subcutaneously administered following surgery. LNP-encapsulated mRNAs Control, VEGF-A, FGF1, or combination of VEGF-A and FGF1, were topically administered on days 0 and 3 of the experiment, using a total volume of 50 µL under a Tegaderm patch each time. The VEGF-A mRNA translates into the protein VEGF165 (Uniprot: P15692-4). The FGF1 mRNA translates into a protein with an N-terminal signal sequence comprising residues 1–21 from human interferon beta (Uniprot: P01574) followed by residues 22–155 from FGF1 (Uniprot: P05230-1) with four punctual mutations: Q40P, S47I, H93G, and C117S, which upon secretion result in FGF1 (22–155) protein with the indicated Q40P, S47I, H93G, and C117S mutations. Peptide sequences of both VEGF-A and FGF1 are detailed in Appendix A. A non-translating mRNA was used as control mRNA (Control) for this study. For the pharmacokinetic studies of individual treatments, the following amounts of mRNA were used (µg): 0.3, 1, and 3. Three µg of Control mRNA were used in the Control group. For the combination treatment, three µg of each mRNA were used, and six µg of Control mRNA were used when the combination of both factors was tested.

### 2.5. Quantification of Human VEGF-A and Human FGF1 Protein Production following Topical Administration

Wounds were created on the backs of the diabetic mice, as described above. VEGF-A mRNA (0.3, 1 or 3 µg) or FGF1 mRNA (1, 3 or 10 µg) formulated in MC3 LNPs were topically administered under the Tegaderm. Mice were anesthetized at predefined time points (6 and 24 h) after topical administration of MC3-VEGF-A mRNA and MC3-FGF1 mRNA. Tissue biopsies from the wound edges (skin) were sampled (50–100 mg), snap frozen in liquid nitrogen, and stored at −80 °C until processed. To evaluate compound distribution to other tissues, biopsies from the liver, spleen, and lung were taken (50–100 mg). A cardiac blood sample was taken for plasma exposure. Tissue and plasma from 3 mice per time point and dose group were analyzed.

Tissue samples were homogenized using a Precellys bead beater system (Bertin Instruments, Montigny le Bretone, France). To each sample, MSD Tris Lysis buffer (R60TX-2, MesoScale Discovery, Rockville, MD, USA) was added at a ratio of 1:10 (*w*/*v*) together with 2.8 mm stainless steel beads (Bertin Instruments, Montigny le Bretone, France). The tissues were homogenized at 6500 rpm for 20 s, four times with 30 s of rest on ice in between runs. The samples were then centrifuged at 10,000× *g* at 4 °C for 10 min and the supernatant was transferred to a new vial and stored at −80 °C pending analysis.

The human VEGF-A protein was quantified using an electrochemiluminescent immunoassay, V-PLEX Human VEGF assay kit (K151RHD, MesoScale Discovery, Rockville, MD, USA). Human FGF1 was measured using two commercial kits, one from R&D Systems and another from Thermo Fisher Scientific (Waltham, MA, USA). The development of a specific immunoassay was also attempted using Gyroslab^®^ (Uppsala, Sweden) platform. The tissue samples were diluted 1:10 in MSD diluent 43 prior to analysis. The standard curve was prepared in MSD43 diluent (MesoScale Discovery, Rockville, MD, USA). Quality control samples were prepared in blank tissue homogenate and diluted like the samples. The plates were read on a MSD Sector Imager 6000 (MesoScale Discovery, Rockville, MD, USA).

### 2.6. Wound Area Measurement

Photographs of the wounds were taken on days 0, 3, 7, 10, 14, and 17 with a Canon camera (EOS 600D, Canon, Tokyo, Japan) at a fixed distance from the wound. The tegaderm was removed and replaced after each examination. The open wound area, identified as the region in the center of the wound lacking an epithelial layer, was measured by tracing the border of the wound in ImageJ2 (NIH, WI, USA). Percent open wound area was calculated by dividing the wound area at each time point by the initial wound area (at day 0) for individual animals.

### 2.7. Evaluation of Angiogenesis

Seven days after the wound was created, the diabetic mice were anesthetized with a 2% inhalable isoflurane (Piramal Critical Care)/oxygen mixture. The rib cage was surgically opened, and the sternum lifted away to expose the heart. A 20 G needle was inserted in the left ventricle of the heart, and the blood was flushed out by the heparin solution (RT) using a perfusion pump (1.5 mL/min). The right atrium was immediately cut open. The mice were then perfused with 8 mL MICROFIL^®^ mixture containing contrast agent (3 mL), MV diluent (6 mL), and a curing agent (0.45 mL), at the same perfusion rate at RT using a 10 mL syringe attached to a 20 G needle via a PE-90 polyethylene tube. After perfusion, the mice were kept at RT for 90 min for the MICROFIL^®^ mixture to polymerize and solidify. The wound and the surrounding skin were excised and stored in 4% formalin, then embedded in blocks with paraffin, separated with paraffin-soaked balsa wood in between. The blocks were horizontally mounted on the CT holder. Micro-computed tomography (CT) was performed using a high-resolution Skyscan 1272 (Bruker, Billerica, MA, USA), 50 kVp, 200 µA, 2700 ms, 3600 projections, and 30 pixel random movement between projections. The projections were reconstructed into a 5 µm × 5 µm × 5 µm CT volume. Raw data were cropped, converted to DICOM files for further image processing and analyzed using Amira Software (2021.2, Berlin, Germany). The vessels were segmented using a fixed threshold, and the total vessel volume was calculated.

### 2.8. RNA Sequencing

RNA sequencing (RNA-seq) of tissue biopsies from diabetic wounds in different healing stages was performed to evaluate the effects of the different treatments (MC3-Control mRNA, MC3-VEGF-A mRNA, MC3-FGF1 mRNA, and the combination of MC3-VEGF-A + FGF1 mRNA) on a molecular level. Biopsies of about 10 mg were collected from the wound edge on days 3 and 7 for RNA extraction using Norgen total RNA extraction kit, including DNase treatment. The RNA integrity was assessed using Fragment Analyzer (Agilent, Santa Clara, CA, USA). RNA concentrations were determined by spectrophotometry (Dropsense/Lunatic, Unchained Labs, Pleasanton, CA, USA), and the final concentrations were normalized to 6.25 ng/μL. To remove any ribosomal RNA (rRNA), the samples were treated with RiboCop rRNA depletion kit for Human/Mouse/Rat (Lexogen, Vienna, Austria). Libraries were generated using the CORALL Total RNA-Seq Library Prep Kit with 12 nt of unique dual indexing (Lexogen). The quality of all amplified libraries was controlled by capillary gel electrophoresis (Fragment Analyzer, Agilent). The sequence-ready libraries were normalized to a concentration of 1.5 nM and pooled, adding a 2% PhiX spike-in. The whole pool was pair-end sequenced (2 × 150 bp) twice on S1 flow cell using S1 Reagent Kit v1.5, 300 cycles according to the NovaSeq XP workflow in the NovaSeq 6000 Sequencing System Guide (Illumina, San Diego, CA, USA). Raw data fastq files were created using bcl2fastq v2.20.0.422. Quality control was performed using MultiQC v1.12. The raw data were processed using the nfcore RNAseq processing pipeline (https://github.com/nf-core/rnaseq, accessed on 8 January 2024). Sequenced reads were quality controlled with the FastQC software and pre-processed with Trim Galore. Processed reads were aligned to the reference genome of *Homo sapiens* (build GRCh38) with the STAR aligner. Read counts for genes were generated using the feature Counts library, and raw gene read counts were generated by the Salmon software (v1.10.0, Dublin, Ireland).

### 2.9. Differential Expression Analysis

The raw gene count data, including 45,706 transcripts from 23 samples, were imported into R for bioinformatic analysis, and statistical testing for differential expression was carried out using the *DESeq2* R package [35]. Filtering and normalization of the raw counts were performed with the *DESeq()* function. The Wald test was used for identification of significantly differentially expressed genes (DEGs) using the following comparisons: VEGF-A vs. Control, FGF1 vs. Control, and VEGF-A + FGF1 vs. Control *p*-values were adjusted for multiple testing using the Benjamin Hoch method, and a combined criteria of false discovery rate (FDR) ≤ 0.05 and log2 fold change (FC) ≥ 1 was considered statistically significant. Standard functions in R were used for exploratory analysis, and the main source of transcriptional variation in the dataset was investigated using principal component analysis (PCA). The number of overlapping DEGs between the different treatments was identified using the VennDiagram R package [36]. The identified significantly enriched DEGs were visualized using functions in the EnhanchedVolcano R-package (https://bioconductor.org/EnhancedVolcano, accessed on 8 January 2024) significantly enriched pathways, biological functions, and upstream regulators in treatment versus control groups. The significance of the enrichment was assessed using Fisher’s exact test with corrected *p*-value < 0.05. An activation Z-score was used to predict the directionality of the regulation; Z > 2 means activated, and Z < −2 means inhibited function/upstream regulator. The Complex Heatmap R-package was used to visualize expression levels of significant DEGs [37].

### 2.10. Statistics

In vitro and in vivo experimental data are expressed as mean ± standard error of the mean (SEM). Statistical analysis for in vitro data was performed with one-way ANOVA and Tukey’s multiple comparisons test using Graphpad Prism 9.5.1. The number of animals per group for the in vivo wound healing model in db/db mice with post-hoc contrasts between treatments of the estimated means at each timepoint made using a pre-defined R package (https://cran.r-project.org/package=emmeans, accessed on 23 January 2024) [38,39]. These models are parameterized with coefficients for the steepness of the response curve, the lower asymptote or limit of the response, and the point at which 50% of the effect is attained, and they are compared between groups. The statistical analysis used for each model is further detailed in each figure legend.

## 3. Results

### 3.1. Punctual Mutations in FGF1 Lead to an Engineered Thermally Stable Protein Variant with an Increased Microvessel Sprouting Induction Capacity

Wild-type FGF1 is relatively unstable, partially unfolded at physiological temperature with a denaturation temperature of 40 °C [23]. FGF1 has been reported to have a short in vivo half-life of 4.8–10.2 min [40]. However, more stable engineered FGF1 variants have been generated that have prolonged biological activity [23,24,25], and therefore, we designed three FGF1 recombinant protein variants and tested them in vitro to be able to identify the best candidate to take forward to mRNA production and subsequent experiments in mice. We expressed and purified (1) wild-type FGF1 (FGF1 (1–155) WT), (2) FGF1 with Q40P, S47I, H93G, and C117S mutations (FGF1 (1–155) MUT), and (3) one N-terminal truncation variant starting at residue 22 with the same four punctual mutations (FGF1 (22–155) MUT). Engineered protein sequences and structures, together with their molecular weight and length, are shown in Figure 1A, Table 1, and Appendix A. Both FGF1 (1–155) WT and FGF1 (1–155) MUT engineered proteins showed higher protein melting temperatures (Tms) of 69.3 ± 0.01 °C and 66.4 ± 3.9 °C, respectively, compared with the wild-type protein (Tm = 50.9 ± 0.04 °C, *p* < 0.01) (Table 1 and Appendix A), indicating increased stability. When tested in an ex-vivo model of microvessel sprouting in adult mice aortic rings (Figure 1B,C), both FGF1 (1–155) MUT and FGF1 (22–155) MUT engineered proteins showed a moderate increase in functional activity compared with the Control group when 100 ng/mL of protein was administered (*p* < 0.05 and *p* < 0.01, respectively). Of important note, a significant increase in sprouting area was observed when samples were treated with FGF1 (22–155) MUT at 100 ng/mL compared with the experimental group treated with the same dose of FGF1 (1–155) WT (*p* < 0.05). Additional images for all experimental conditions are provided in Appendix A. FGF1 is secreted through an unconventional protein secretion pathway induced by stress conditions like starvation and heat shock [41]. However, a previous study performed by J. Jouanneau and colleagues demonstrated that the addition of a classical signal sequence to human acidic FGF1 induces the secretion of functional FGF1 into the media through the classical secretion pathway [26]. This finding, together with our data, supported the idea of using the FGF1 (22–155) MUT sequence to synthesize the mRNA used for the following studies, which would be referred to from now on as FGF1.

### 3.2. Combination of VEGF-A and FGF1 Proteins Shows an Additive Effect on Endothelial Microvessel Sprouting

To investigate the potential additive effect of VEGF-A and FGF1 on microvessel sprouting, adult mouse aortic rings were treated with VEGF-A and FGF1 (22–155) MUT (FGF1 from now on) individually and in combination for 7 days. As shown, in Appendix A, the combination of VEGF-A (10 ng/mL) and FGF1 (25 and 50 ng/mL) already induced significant endothelial microvessel formation at day 5 compared with Control (*p* < 0.05). Although a trend of increased sprouting area was observed when aortic rings were treated with the combination of VEGF-A and FGF1 at different doses, no statistical significance was observed when compared with the individual treatments (VEGF-A 10 ng/mL, FGF1 25 ng/mL, and FGF1 50 mg/mL). In contrast, differences between experimental groups were enhanced after 7 days of treatment (Figure 1D,E and Appendix A). The sprouting area of aortic rings treated with C1 and C2 was higher than that of the Control group (*p* < 0.00001 and *p* < 0.001 for C1 and C2, respectively) and individual treatments. No statistically significant differences were observed when the Control group was compared with individual treatments, indicating an additive effect of the VEGF-A and FGF1 combination on angiogenesis enhancement in aortic rings ex vivo.

### 3.3. Human VEGF-A (h-VEGF-A) Protein Is Detected in the Wound Area in a mRNA-Hose-Dependent Manner

A set of animals was used to evaluate the protein production from VEGF-A mRNAs in diabetic mice after 6 and 24 h of surgeries and topical administration, where three animals per experimental condition were included. Increasing doses of VEGF-A mRNA were tested (µg): 0.3, 1, and 3. Topical administration of VEGF-A mRNA in the wound resulted in robust protein production at both 6 and 24 h post-injection. There was a clear relationship between the encapsulated mRNA dose and the hVEGF-A protein amount detected, where the highest mRNA dose also led to the highest protein production (Figure 2A). hVEGF-A protein was detected in all skin samples except two in the lowest dose group at 24 h after dosing (Figure 2A). There was low exposure in plasma after 6 h of 3 µg VEGF-A mRNA administration, while there were no detectable levels in the other dose groups or time points (Figure 2B). Minor levels of hVEGF-A protein were found in the liver and spleen at 6 h and 24 h after administration. hVEGF was not detected either in lung samples in any of the experimental conditions or in the spleen after 6 and 24 h of mRNA administration in all dose groups (Figure 2C–E). Human FGF1 protein concentration was not possible to measure due to technical assay limitations since human FGF1 produced from the mRNA could not be distinguished from rodent endogenous FGF1.

### 3.4. Monotherapy of VEGF-A mRNA and FGF1 mRNA, Respectively, Improves the Wound Healing Rate in a Dose-Dependent Manner

We evaluated the dose-response of the wound healing process in diabetic mice when either VEGF-A mRNA or FGF1 mRNA were topically administered as monotherapy at days 0 and 3 post-surgery, using increasing doses (µg: 0.1, 0.3, 1, and 3). On day 7 post-surgery, the animals treated with 1 and 3 µg VEGF-A mRNA exhibited significantly smaller open-wound areas versus the Control mRNA group (21.6 ± 6.3% and 7.1 ± 1.1% vs. 53.1 ± 7.6%, respectively). On day 10, the 3 µg VEGF-A mRNA-treated group (0.5 ± 0.3%) showed a significantly smaller open-wound area versus the Control mRNA group (32.4 ± 6.7%) (Figure 3A). Similar effects were observed when animals were treated with FGF1 mRNA. On day 7 post-surgery, the group treated with 3 µg FGF1 mRNA (16.05 ± 2.95%) and the group treated with 10 µg FGF1 mRNA (4.1 ± 1.3%) showed significantly smaller open-wound area versus the Control mRNA-treated group (44.2 ± 5.9%). On day 10, the group treated with 10 µg FGF1 mRNA displayed a significantly smaller open-wound area versus the control group (0.24 ± 0.24% vs. 9 ± 2.9%, respectively) (Figure 3B). These results pointed to an increased wound healing rate in a dose-dependent manner.

### 3.5. Combination of VEGF-A and FGF1 mRNAs Leads to an Accelerated Wound Healing Rate Compared with Single Treatments

After evaluating the response to a range of mRNA doses of VEGF-A or FGF1 treatments, we selected the dose of 3 µg of mRNA of each compound to test if the combination could increase the wound healing rate in diabetic mice. The wound area was measured over time after topical administration of the treatments on days 0 and 3, and the combination was compared with VEGF-A and FGF1 alone. Results and representative images for all experimental groups are shown in Figure 3C,D and Appendix A. VEGF-A mRNA and FGF1 mRNA monotherapy significantly reduced the wound healing area at day 7 compared with the Control group (wound area (%): 43.9 ± 7.4 vs. 21 ± 7.3 and 20 ± 4, respectively; *p* < 0.05). Importantly, the combination of VEGF-A mRNA + FGF1 mRNA had a significant additive effect on the wound healing rate compared with the control group (wound area (%): 43.9 ± 7.4 vs. 4.7 ± 1.4, *p* < 0.001), as well as VEGF-A mRNA and FGF1mRNA monotherapy (wound area (%): 4.7 ± 1.5 vs. 21 ± 7.3 and 20 ± 4, *p* < 0.05 and *p* < 0.01, respectively) at day 7. Altogether, data showed an additive effect on wound healing rate when a topical administration of VEGF-A mRNA in combination with FGF1 mRNA encapsulated in MC3 LNPs was administered compared with control or monotherapy of VEGF-A or FGF1 mRNA, respectively.

### 3.6. Neovascularization on the Wound Area Is Enhanced with Topical Administration of Combination of Encapsulated VEGF-A and FGF1 mRNAs

Impaired vascularization results in chronic wounds and poor-quality tissue repair [42]. Subsequently, we evaluated the capacity of VEGF-A, FGF1, and the combination to promote neovascularization and quantified the volume of vessel formation by microCT angiography. As shown in Figure 3E,F, we observed a significant increase in vessel volume in the wound area when animals were treated with a VEGF-A + FGF1 mRNA combination (mRNA dose: 3 µg for each compound). In contrast, monotherapies did not significantly increase neovascularization compared with the control group.

### 3.7. Distinct Transcriptional Machinery Is Activated by VEGF-A, FGF1, and VEGF-A+FGF1 mRNA Treatments with Low Overlap of Expression Profiles between Groups

A robust transcriptional response was observed shortly after wound induction, with the most significant variation in gene expression already evident at day 3 post-surgery. PCA analysis incorporating both day 3 and day 7 revealed that 10.1% of the overall variation could be attributed to differences between the time points. A clear separation between day 3 and day 7 was apparent in PC3, accounting for 10.1% of the variance (Figure 4A). The PCA, including all treatments at day 3 (Figure 4B), displayed separation attributed to each treatment. PC1 explained 24.4% of the total variation,and PC2 accounted for 20.1% of the variance. Notably, a distinct separation of the combination treatment with VEGF-A mRNA + FGF1 mRNA compared with control samples was evident (Figure 4B). The transcriptional response was strongest at day 3 and rapidly declined over time, with relatively few DEGs detected 7 days after surgery (Appendix A). Consequently, all subsequent analyses were focused on the transcriptional patterns at day 3.

The most substantial transcriptional changes were observed in the groups treated with FGF1 mRNA and the combination of VEGF-A mRNA + FGF1 mRNA compared with the control group. The number of DEGs identified for each comparison is illustrated in Figure 4C–E and detailed in Appendix A. Comparing the overlap of DEGs among the treatment groups revealed that only 27 genes were differentially expressed in all treatment groups (Figure 4F). This suggests that the combination of VEGF-A mRNA and FGF1 mRNA induces additional transcriptional responses. Notably, the combined treatment exhibited a larger overlap with the FGF1 mRNA treatment (28%) than with the VEGF-A mRNA treatment (12%). Moreover, a total of 107 genes were uniquely differentially expressed when the combination of VEGF-A mRNA+FGF1 mRNA was applied. 

### 3.8. The Combined Treatment of VEGF-A and FGF1 mRNAs Activates Tissue Regeneration-Related Biological Functions

Functional enrichment analysis of the identified DEGs at day 3 unveiled a significant activation of biological functions associated with tissue regeneration. These functions encompassed “development of vasculature”, “angiogenesis”, “vasculogenesis”, “organization of cytoplasm, “sprouting”, “proliferation of endothelial cells”, “growth of epithelial tissue”, and “growth of skin” in the experimental group treated with the combination of VEGF-A and FGF1 mRNAs (Figure 5A). Except for “growth of skin,” all these other biological functions were also significantly activated in the VEGF-A mRNA treatment group. Conversely, among these biological functions, only “sprouting” and “organization of cytoplasm” were significantly activated in the FGF1 group (Figure 5A).

As multiple biological functions are linked to similar biological processes and share a considerable number of DEGs, we selected “angiogenesis”, “growth of epithelial tissue”, and “growth of skin” as key representative processes of the wound healing process and looked in more detail into the expression of the genes associated with these functions in the different treatments. Initially, we performed an upstream regulator analysis and identified four regulators directly or indirectly associated with these biological functions: FN1, SPP1, FGF1, and F2R (Figure 5B). All these upstream regulators were significantly up-regulated in the VEGF-A+FGF1 mRNA treatment group compared with the control group (Figure 5C), suggesting an additive effect in our wound healing model when both mRNAs are used in combination. We also visualized the expression levels of all DEGs that were linked to at least two out of the three investigated biological functions: “angiogenesis”, “growth of epithelial tissue”, and “growth of skin” (Figure 5D). From a total of 69 genes commonly associated with these functions, 10 genes show significant upregulation only in the group where both VEGF-A mRNA and FGF1 mRNA were administered together: MMP3, TIMP1, PAQR3, ODC1, PGF, MDK, FBLN2, and HMGA2 (Figure 5E). Both FN1 and MMP3 were also significantly upregulated in individual treatments, but with stronger significance in the combined treatment. Overall, these findings support that the combined treatment with both VEGF-A mRNA and FGF-1 mRNA has an additive effect on skin regeneration potential in our in vivo wound healing model. This is evident through a distinctive transcriptomic profile and the significant upregulation of key genes known to play a crucial role in the wound healing process.

## 4. Discussion

The unresolved wound healing process in DFU is a multifactorial problem involving inflammation, angiogenesis, and extracellular matrix remodeling in diabetic patients [2]. Revascularization of the ulcerous area is critical to overcome ischemia in the injured tissue [4,5,6,7,8]. Preclinical models have indicated the potential beneficial effects in this context [10,43]. Comprehensive studies of the wound healing process have also demonstrated the importance of additional growth factors in cell repair processes, among which FGF1 has gained special attention [14,44]. This study reveals an additive effect by combining VEGF-A and a more stable variant of FGF1, surpassing the outcomes of monotherapy with VEGF-A or FGF1 alone. The synergistic application of both demonstrated enhanced microvascular sprouting in an ex vivo aortic ring model, along with an increase in microvessel and wound healing rates in diabetic mice. These factors were delivered topically as modified mRNA formulated in LNP in vivo, which is an exciting modality that could deliver the drugs with the optimal PK profile to induce regenerative effects and wound healing.

Previous clinical trials have focused on diverse strategies to treat DFU, including the delivery of topical recombinant growth factors [15,45,46,47,48], platelet-rich plasma [49], umbilical cord mesenchymal stromal cells (MSC) [50], macrophage-regulating drugs (NCT01898923) [51], or enriched hydrogels with sodium alginate and vitamins. The use of mRNA holds significant promise in the exploration of new pharmaceuticals, especially when upregulation of a protein is desired [52]. Importantly, mRNA requires stabilization and protection to prevent degradation once it has reached the step of translation in the target tissue. LNP has been widely used and shows promising effects and acceptable safety [27,28]. In this study, we have used the LNP MC3 as a formulation tool to deliver VEGF-A and FGF1 mRNAs to a wound via topical application. Our findings demonstrate efficient protein formation and effects on wound healing in a way that requires lower doses of mRNA. This administration is more convenient and less resource- and skill-demanding compared with the intradermal injection used in our previously published wound healing study with VEGF mRNA in diabetic mice [18]. In the current study, we did not observe a major leakage of human VEGF-A protein to other tissues after administration, such as the liver, spleen, and lung, neither after 6 nor 24 h of treatment, while it was possible to detect high levels of human VEGF-A protein in the skin samples. We detected VEGF-A in plasma after 6 h of treatment but not after 24 h, indicating a protein washout from the circulation. When translated into clinical practice, a topical formulation with LNP instead of intradermal injection will improve the efficiency of the treatment, decrease the dose needed, and diminish the skill and expertise of health care personnel required to administer the treatment. Hence, topical administration of LNP-formulated mRNA holds the potential to decrease the cost of and enable broader access to mRNA treatment for DFU. This holds true even if there is an additional cost for LNP formulation per se.

In diabetic patients, impaired angiogenesis and neovascularization are two of the multiple factors leading to defective wound healing [53]. Consequently, the finding that a combination of VEGF-A and FGF1 can exert an additive impact on the induction of angiogenesis and vasculogenesis in the wound healing area compared with the monotherapies holds promise as a therapeutic approach. In addition, the combined treatment resulted in a significant upregulation in the expression of genes involved in angiogenesis, the development of the vasculature, and vasculogenesis, such as HMGA2 [54], F2R [55], and PGF [56], compared with the individual treatments 3 days post-surgery. Furthermore, PAQR3 downregulation in this group correlates with previous findings, where it was demonstrated that its depletion accelerates wound healing by promoting angiogenesis [57,58]. Since previous studies have shown that a HMGA2 response can be triggered upon FGF1 stimulation [59], an enhanced expression when using the combination treatment in our study supports the idea of this concept. Interestingly, both metalloproteinases 3 (MMP3) and 9 (MMP9) and their up-regulators fibronectin (FN1) and tissue inhibitor metalloproteinase 1 (TIMP1) were found to be significantly upregulated in the samples treated with the FGF1 and VEGF-A mRNA combinations. These factors have been identified as early responders to tissue injury, actively regulating the inflammatory phase of healing by regulating ECM degradation, stimulating leukocyte infiltration for resolution of the inflammation phase, and transitioning to the proliferative phase [60,61,62,63]. Moreover, FN1 plays a role in recruiting endothelial cells and fibroblasts into the wound region, promoting the migration of epidermal cells [64]. The upregulation of ODC1, MDK, FBLN2, and SPP1 in samples from diabetic mice treated with the combined treatment also reinforces the idea that combining these factors for DFU treatment can enhance cell migration in chronic wounds, since they have also been previously linked to a reparative response in this in vivo model promoting epidermal regeneration and acute inflammatory response [65,66,67,68,69].

The fact that the strongest transcriptional response was observed at day 3 of the wound healing process is well aligned with the dynamics of growth factors involved in the wound healing process, where there is an early proliferative phase characterized by re-epithelialization, angiogenesis, and the formation of granulation tissue that leads to closure of the epithelial layer, revascularization in the damaged area, and tissue regeneration [70,71]. VEGF-A has been shown to have a rapid increase during the first five days of wound healing [72]. This could explain the rather low number of differentially expressed genes at a later phase (day 7). However, this could also be a consequence of the dosing schedule since the treatments were added on days 0 and 3 after surgeries. Although these administration time points have demonstrated regeneration potential in our in vivo model, further in vivo studies including additional dosing time points and concentrations of the combined treatment would be interesting to explore for an improved wound healing process. A second limitation of this study was that the levels of human FGF1 protein could not be measured in the analyzed tissues. We were unable to differentiate between the human and endogenous mouse proteins after investigating different commercial kits as well as a platform specifically developed for this purpose. Therefore, the question remains regarding the PK profile and biodistribution of FGF1 mRNA. A third limitation of this study is related to the deviation of the transcriptional profile of two of the FGF1 mRNA-treated samples from the rest of the samples in that group. These samples showed similarity to the control group in the clustering analysis. This could be a consequence of either biological variability or technical problems with the administration, leading to a lack of response in two animals in the experimental group. Since we failed to find evidence for any technical issues and could not establish the source of this variation, we decided to include the samples in the analysis. However, care should be taken with any conclusion drawn from these samples.

## 5. Conclusions

Altogether, our findings demonstrate an additive effect resulting from the combination of VEGF-A and FGF1 recombinant proteins on angiogenesis, as evidenced by enhanced tube formation, microvessel sprouting, and neovascularization. Additionally, we have also shown that a topically delivered LNP-formulated combination of VEGF-A and FGF1 mRNAs improves wound closure in a diabetic mouse model of wound healing compared with monotherapies, thereby supporting a more accessible administration method from a clinical perspective. Importantly, transcriptomic analysis of mouse skin wound samples revealed a significant upregulation of genes linked to skin growth, angiogenesis, and epithelial cell proliferation compared with untreated samples or samples exposed to single growth factor treatments. This supports the hypothesis that a combinatorial approach based on VEGF-A and engineered FGF-1 enhances regenerative properties. While further validation and extensive studies are necessary, our data endorse further investigations of LNP-encapsulated VEGF-A and FGF1 mRNA combinations as a potential future treatment for DFU.

## Figures and Tables

**Figure 1 cells-13-00414-f001:**
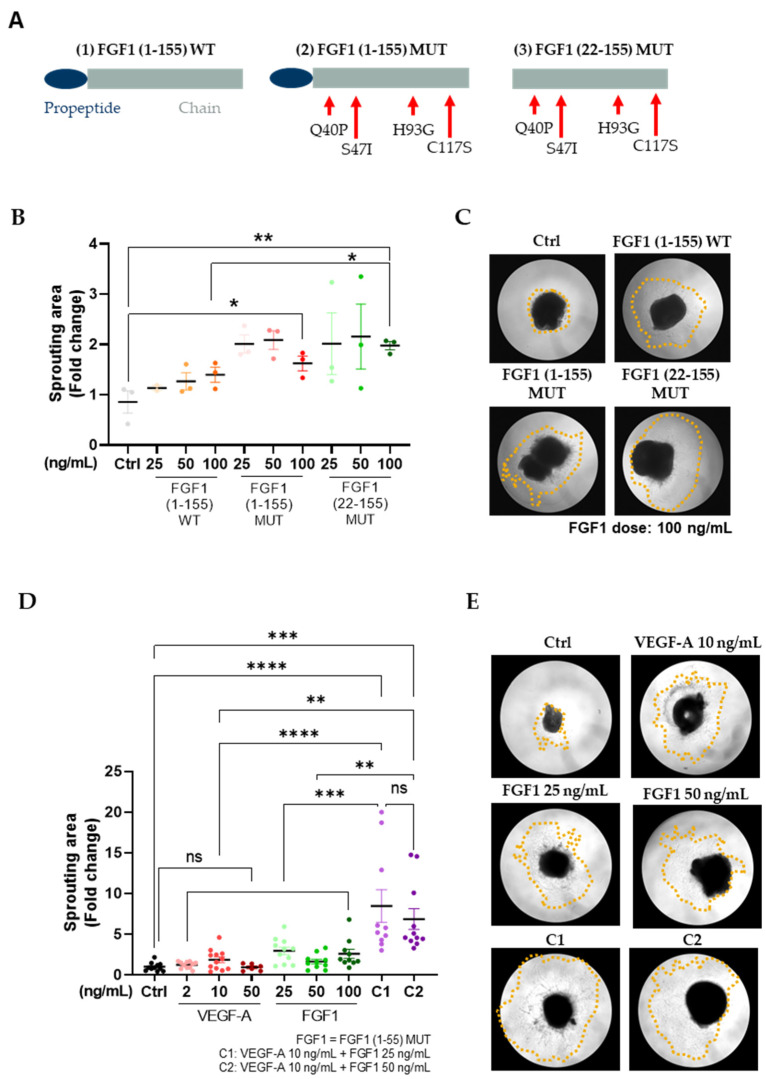
Combination of VEGF-A and FGF1 (1–155) MUT recombinant proteins shows increased microvessel sprouting in an aortic ring ex vivo model. (**A**) Schematic representation of the FGF1 recombinant proteins. (1) FGF1 (1–155) WT: wild-type FGF1, containing the full native sequence; (2) FGF1 (1–155) MUT: FGF1 full sequence where point mutations have been introduced in the specified positions; (3) FGF1 (22–155): FGF1 protein where propeptide domain has been removed. The same point mutations as before have been introduced. Propeptide domain is shown in blue, and main chain is shown in grey. Q = Glutamine; P = Proline; S = Serine; I = Isoleucine; H = Histidine; G = Glycine; C = Cysteine. (**B**,**C**) Sprouting area and representative images of FGF1 variants. Images show aortic rings treated with FGF1 variants at 100 ng/mL. (**D**,**E**) Microvessel occupied area when aortic rings treated with VEGF-A, FGF1 (1–155), MUT (FGF1), and a combination of VEGF-A and FGF1 at different concentrations. Representative pictures of VEGF-A, FGF1, and the combination of VEG-A and FGF1 after 7 days of treatment are shown. An equivalent volume of diluent (PBS) was added to control samples (Control). Treatments were refreshed every 48 h. Yellow dashed lines represent the area of microvessel sprouting. Data show microvessel occupied area at day 7, normalized to Control values. Data have been collected from 12 adult mice in three independent experiments. Mean ± SEM and individual values are plotted. One-way ANOVA followed by Tukey´s multiple comparison test has been used for statistical analysis. Differences between experimental groups were considered significant at *p* < 0.05 (ns: non-significant; * *p* < 0.05; ** *p* < 0.01; *** *p* < 0.001; **** *p* < 0.0001).

**Figure 2 cells-13-00414-f002:**
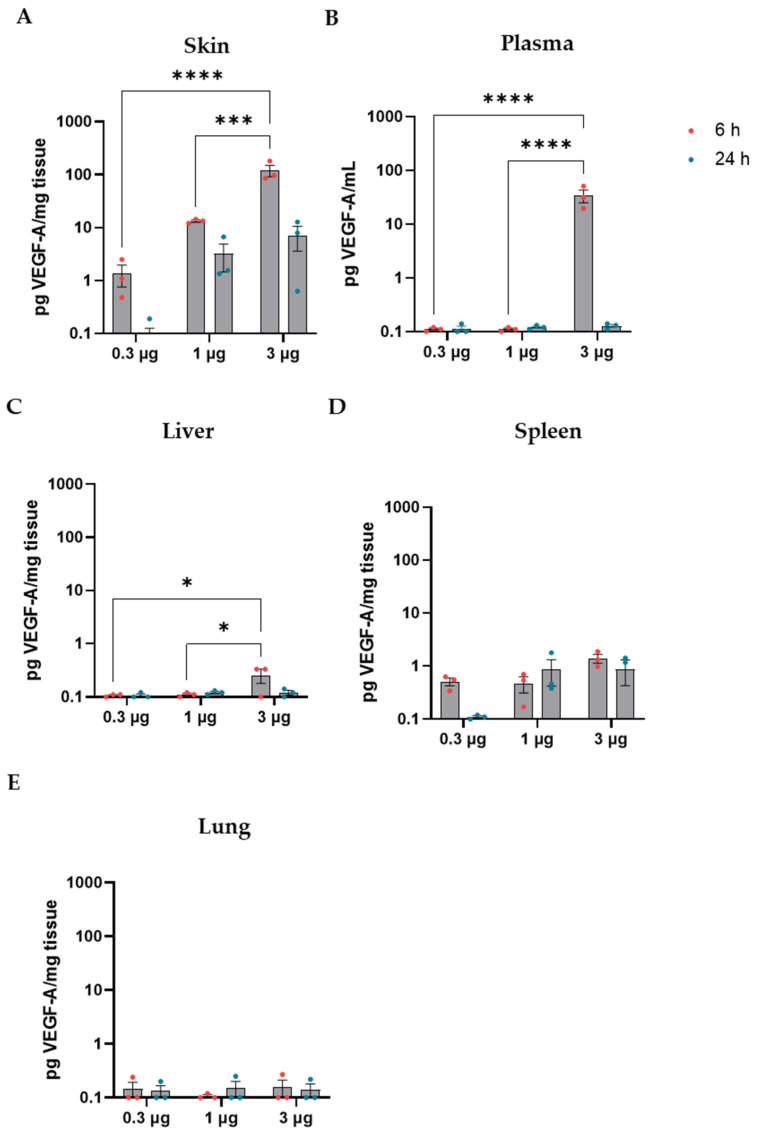
Human VEGF-A protein measurement in tissue and plasma samples after VEGF-A mRNA administration in vivo. Three doses of mRNA were administered (µg: 0.3, 1 and 3) and VEGF-A protein was measured on (**A**) Skin, (**B**) Plasma, (**C**) Liver, (**D**) Spleen, and (**E**) Lung. Samples were taken 6 and 24 h after VEGF-A topical administration to the skin. Data are represented as mean ± SEM in a log scale as pg of VEGF-A per mg of tissue sample from three animals from each experimental group. Two-way ANOVA was used to evaluate statistically significant differences, considered when *p* < 0.05 (* *p* < 0.05; *** *p* < 0.001; **** *p* < 0.0001).

**Figure 3 cells-13-00414-f003:**
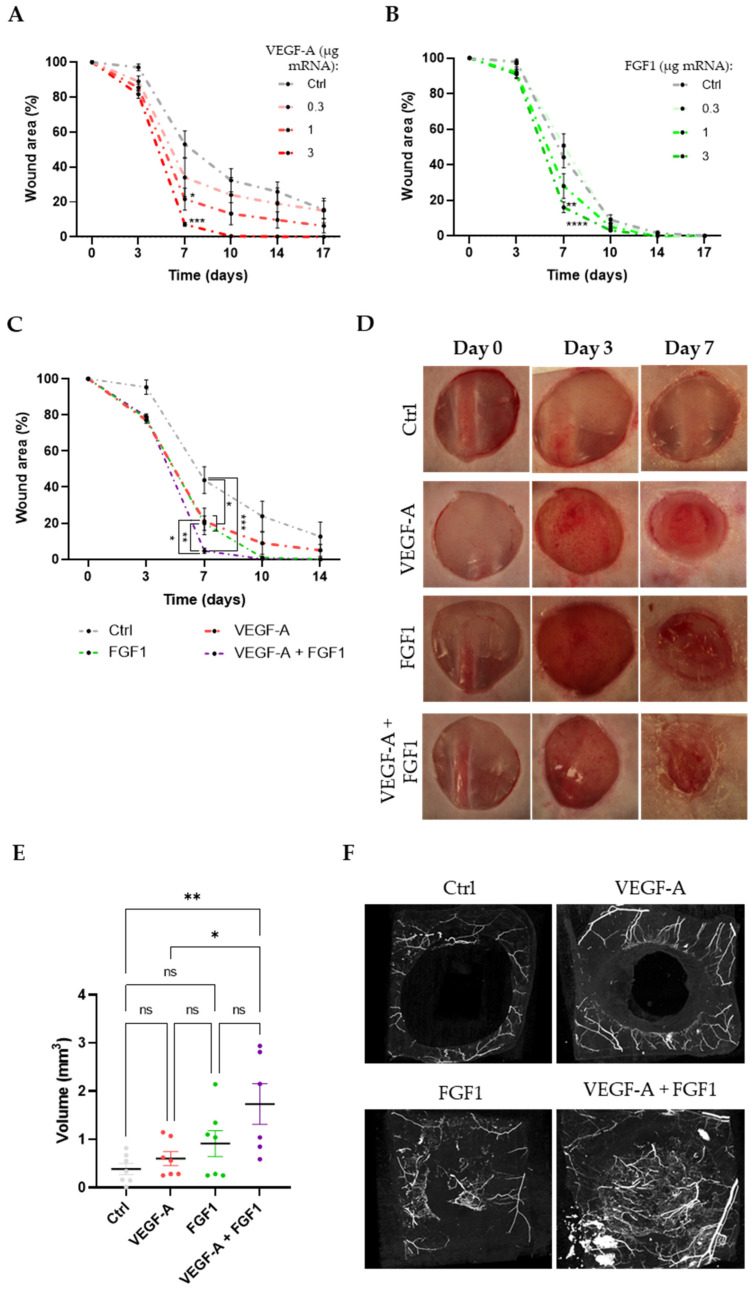
Topical administration of VEGF-A and FGF1 mRNA in combination shows an additive effect on wound healing in diabetic mice. Wound area after treatment with different doses of (**A**) VEGF-A or (**B**) FGF1 mRNA encapsulated in MC3 LNPs over time. Dose-encapsulated mRNA (µg): 0.3, 1, 3. (**C**) Wound area measurement over time on animals treated with VEGF-A mRNA, FGF1 mRNA, or VEGF-A+FGF1 mRNA (encapsulated mRNA: 3 µg for monotherapy and 6 µg for Control). (**D**) Representative images of wound healing progression for each experimental group at days 0, 3, and 7. A non-coding mRNA was used as control (Control, mRNA dose: 3 µg). Dosing was administered on days 0 and 3. Wound area was measured over time and calculated using the wound area on day 0 as reference value. (**E**) Calculated vessel volume (mm^3^) on the wound area using micro-CT images with an X-ray dense casting after treatment with 3 µg of each mRNA. (**F**) Examples of 3D reconstruction of micro-CT images showing vasculature in the wound area on day 7. A total of seven animals were included in each experimental group. Data are presented as mean ± SEM. One-way ANOVA followed by Tukey´s multiple comparison test was used to compare treatments at each time point. Differences between experimental groups were considered significant at *p* < 0.05 (ns: non-significant; * *p* < 0.05; ** *p* < 0.01; *** *p* < 0.001; **** *p* < 0.0001).

**Figure 4 cells-13-00414-f004:**
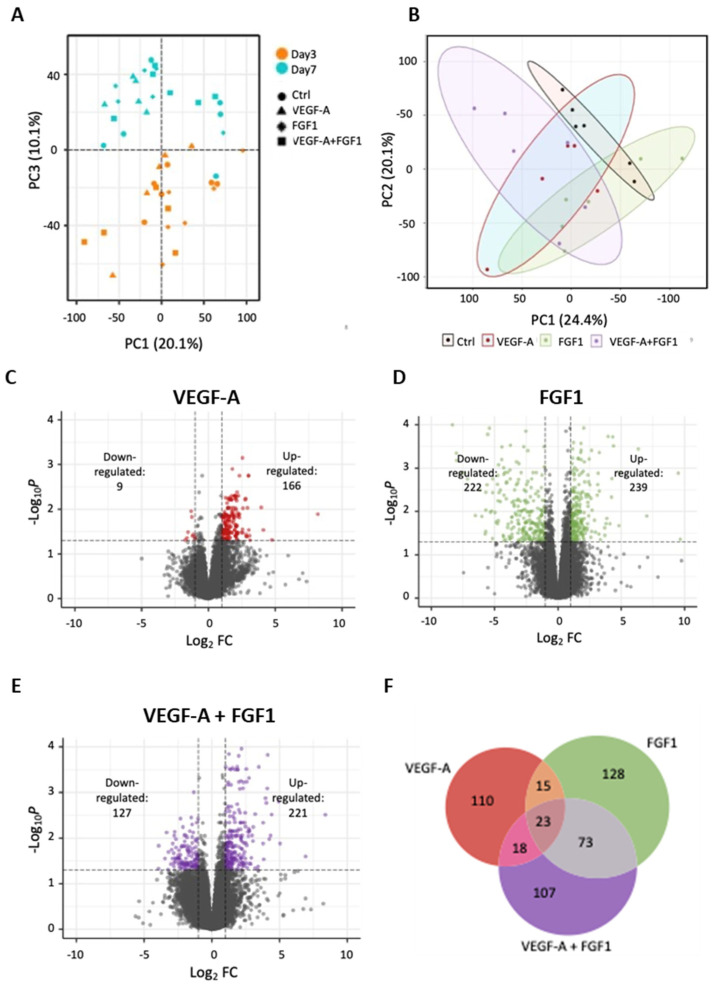
RNA-seq analysis reveals a strong differential gene expression at day 3 of the wound healing process. (**A**) Principal Component Analysis (PCA) for the four experimental groups at days 3 and 7 of the wound healing process. The first component (PC1) accounts for 20.1%, and the third component (PC3) accounts for 10.1% of the overall variance observed in the data. (**B**) PCA plot showing the variance between treatment and control groups at day 3. The first two components account for 24.4% and 20.1% of the variance, respectively. (**C**–**E**) Volcano plots of differentially expressed genes between treatments (both as monotherapies and in combination) and the control group. Colored dots indicate significantly differentially expressed genes for each experimental group (log_2_ fold change > |1| and adjusted *p*-value < 0.05). (**F**) Venn diagram showing the number of significantly differentially expressed genes that are either unique for each treatment or overlapping between the three treatments.

**Figure 5 cells-13-00414-f005:**
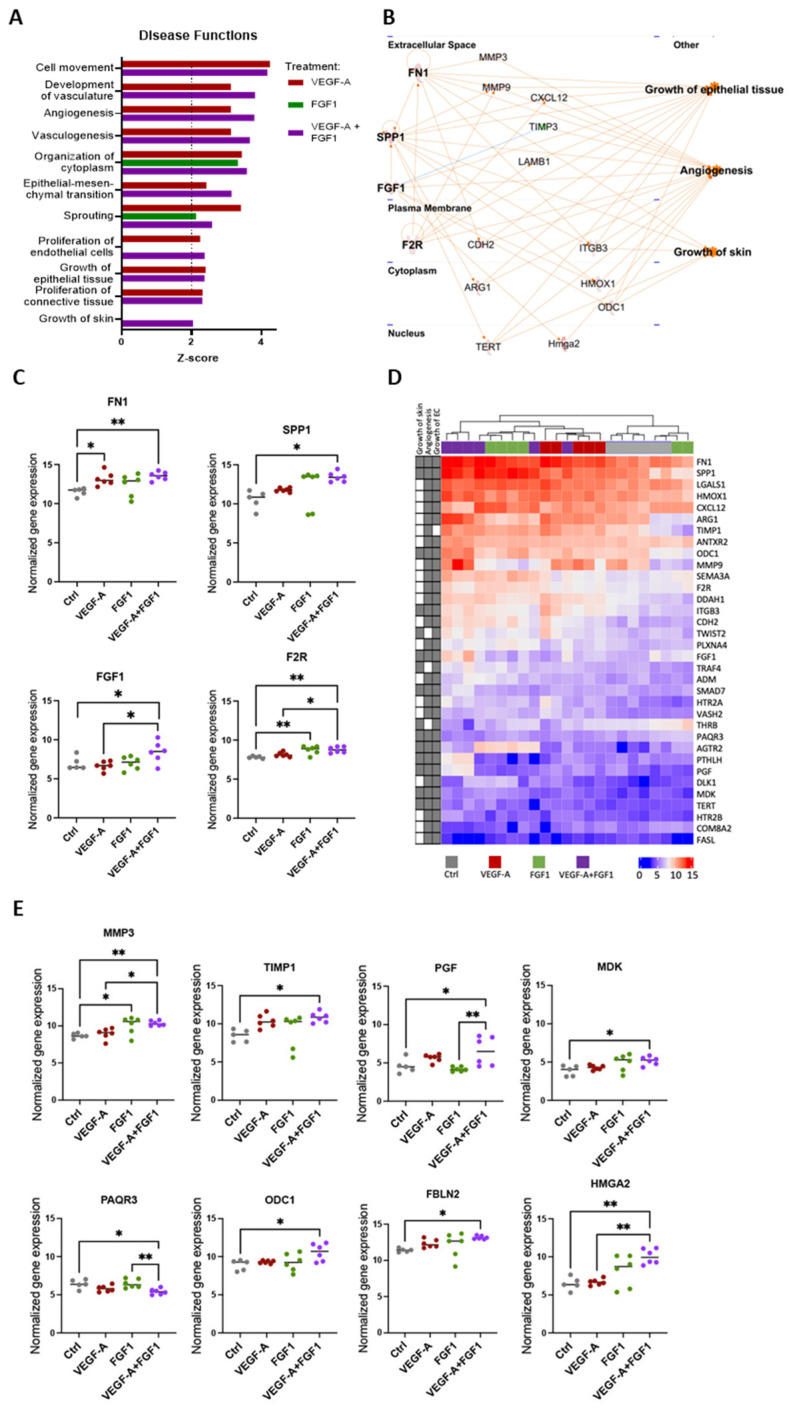
Combination of VEGF-A+FGF1 mRNAs demonstrates enhanced activation of regeneration-associated biological pathways. (**A**) Significantly up-regulated functions (Z-score > 2) of relevance for wound healing and regeneration were identified and explored in more detail in the individual and the combined treatments using IPA. (**B**) Network analysis of genes involved in three disease functions associated with wound healing that showed significant upregulation in the combined treatment. Four up-stream regulators (FN1, SPP1, FGF1, and F2R) for these disease functions were identified using IPA. (**C**) Gene expression levels of the upstream regulators identified from the network analysis. (**D**) Heatmap clustering of samples at day 3 post-surgery. The heatmap illustrates the gene expression for all genes that are involved in at least two of the three selected disease functions (growth of skin, angiogenesis, and growth of epithelial tissue), indicated with gray squares on the left side of the heatmap. Color scale shows normalized gene expression levels. (**E**) Expression values of a selection of genes that show the most significant changes in the combined treatment. One-way ANOVA followed by Tukey’s multiple comparison test was used to compare treatments at each time point. Differences between experimental groups were considered significant at *p* < 0.05 (* *p* < 0.05; ** *p* < 0.01).

**Table 1 cells-13-00414-t001:** FGF1 recombinant protein characteristics.

Parameters	FGF1 (1–155) WT	FGF1 (1–155) MUT	FGF1 (22–155) MUT
Mw ^1^ (g/mol)	17,516.75	17,415.67	15,166.11
Mw (kDa), aprox	17.52	17.42	15.11
Length (aa ^2^)	155	155	133
Tm ^3^ (mean ± SEM, °C)	50.94 ± 0.05	69.38 ± 0.01	66.41 ± 3.9

^1^ Mw: Molecular weight; ^2^ aa = amino acids; ^3^ Tm: melting temperature.

## Data Availability

The transcriptomics dataset generated and presented in this study are openly available in FigShare at https://figshare.com/s/ebd45c896a3daad2a364 with DOI: 10.6084/m9.figshare.24960894, containing raw counts, normalized TPM values, and metadata with a description of the experimental setup.

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
