# Peer review of "The Combination of Vascular Endothelial Growth Factor A (VEGF-A) and Fibroblast Growth Factor 1 (FGF1) Modified mRNA Improves Wound Healing in Diabetic Mice: An Ex Vivo and In Vivo Investigation"

_cells, 2024, doi:10.3390/cells13050414_

Round 1

Reviewer 1 Report

Comments and Suggestions for Authors

The authors have conducted this ex vivo / in vivo study to investigate the outcome differences of lipid nanoparticles (LNP) encapsulated VEGF-A, FGF1 and their combination in topical application for wound healing purposes. The authors’ main purpose with this original investigation is to find out which of these 3 groups has the best outcomes regarding their angiogenesis and revascularization inducing capabilities. They have concluded that the combination of VEGF-A with FGF1 had the most promising outcomes and could possibly be even considered for human clinical trials.

While I find the topic of this paper to be very interesting and on par with the trend of regenerative medicine in wound healing procedures/treatments, I still came across a lot of issues listed below:

Title:

I advise the authors add the term “The” at the start of their title.

 Also, I suggest applying the following changes:

The Combination of vascular endothelial growth factor A (VEGF-A)

and fibroblast growth factor 1 (FGF1)-modified mRNA improves wound healing in diabetic mice: An ex vivo and in vivo investigation

Abstract:

The authors have used numerical characters for each section. Kindly remove those numbers. 

Line 25:

Remain and not “remaining”. Highly ant not “high”.

Line 33:

“7 days after surgery” after which surgery? The authors must directly refer to their surgical procedure, was it an anastomosis or revascularization procedure?

Lines 34 and 35:

“wound healing process at day 3 after surgeries,” once again, the authors must clarify which surgeries they are speaking of?

Lines 36 and 37:

“Combination of LNP-encapsulated VEGF-A and FGF1 mRNAs is a promising approach to improve the scarring process in DFU.” The authors have used the abbreviation term “LNP” (as in lipid nanoparticles), without giving the full form of the term anywhere in their abstract. If you want to use abbreviations, you must write it in full form first and from then on you can use it in its abbreviated form.

Keywords:

I believe it would be beneficial to add the terms “diabetes”, “angiogenesis” and “revascularization” to your list of keywords. 

Introduction:

Overall, the whole manuscript, specifically the abstract, introduction and discussion need serious revisions and reconstructions regarding their English grammar and choices of words. I highly advise the authors take aid from online English-grammar-checking programs to significantly improve their manuscript. The authors can also take aid from the MDPI online English-grammar checking tool. 

Lines 100, 101, 102, 103, 104, and 105:

“Interestingly, our results show faster wound healing effects when 100

 combining VEGF-A and FGF1 mRNAs. Transcriptomic analysis of mice skin wound samples showed significant upregulation of genes linked to growth of skin, angiogenesis and proliferation of epithelial cells compared to untreated samples or samples that were exposed to single growth factor treatments, supporting the hypothesis that a combinatorial approach based on VEGF-A and engineered FGF-1 increases the regenerative properties.” The last paragraph of an introduction in an original study must only focus on the research gaps and questions raised in the authors’ minds after their extensive electronic search, and their main study goals and purposes and even their hypotheses and possible answers to their research questions. Other than these data, the authors must not include their results or conclusions here. But here, the authors have fully dedicated their last paragraph to their results and conclusions which in my opinion is not appropriate. 

The references used for each section of the introduction are all relevant and are all relatively newly-published. I am quite pleased with the authors’ efforts to include the most relevant studies they could have had found. In addition, most of the epidemiologic statements and claims of the authors have been supported by long term, numerous-patient supported data published in legitimate journals, which has immensely improved the reliability of their statements.

Methods and Materials:

Lines 133, 134, and 135:

“Thoracic aortae were first dissected from C57BL/6 female adult mice (from 133 20 to 22 weeks old) and a set of forceps were used to remove the fatty layer. Then, each 134 arteria was cut into 3 to 5 mm rings.” The authors have described these surgical steps and have also referred to the original study that they have borrowed this technique from. However, in the “Ex vivo aortic ring assay” section, ethical committee approvals for these surgical procedures on animal models are nowhere to be found. On the other hand, in the “Wound healing in vivo model” section, the authors have mentioned the ethical approval that they obtained for their in vivo investigations and surgical procedures. So my question is, did the authors use that same ethical approval for their ex vivo investigations or did they obtain a completely different one? Either way, the authors must be more transparent about their ethical approvals and must definitely mention their committee approvals in all sections of their methods that is dedicated to surgical procedures on animal models whether is in an in vivo or ex vivo context.

Lines 177-181:

“A total 13 animals 177 per group were included in the study. Wound healing area and angiogenesis experiments 178 were measured in 7 animals per group, while protein expression levels at two different 179 time-points as measured in 6 animals per group, obtaining data from 3 animals for each 180 time point.” The authors need to fully explain to their readers as to why all 13 animals in each group were not investigated for the same tests. Also if the authors have followed a certain protocol or study with a similar research strategy they must definitely refer to it. 

Lines 183-a85:

“A 1 cm in diameter full thickness skin wound 183 (including dermis and epidermis) was surgically made on the dorsum of the mouse and 184 the wounds were covered with a Tegaderm dressing.” As I mentioned in the abstract section, the authors need to clarify what kind of surgery these animal models underwent in their abstract.

Results:

All of the different figures and charts are all high in quality and easy to comprehend. The authors have successfully determined their investigations in full details except for Figure 5. The small graphics in A, B, C, and E sections can all benefit from a quality upgrade. Kindly revise these graphs and figures. 

And my final note would be as to why the authors chose to not have a conclusion section in their manuscript? As I mentioned in the introduction section, the authors had put their main results and conclusions at the end of their introduction. I advise removing those conclusions from your introduction and have a separate conclusions section in your manuscript after your discussion. 

“Altogether, we have demonstrated an additive effect of a combination of VEGF-A 626 and FGF1 recombinant proteins on angiogenesis in terms of tube formation, microvessel 627 sprouting and neovascularization. We have also shown that a topically delivered LNP- 628 formulated combination of VEGF-A and FGF1 mRNAs improves wound closure in a dia- 629 betic mice model of wound healing, supporting an easier administration method. Alt- 630 hough further validation and more extensive studies will be needed, our current data sup- 631 port further investigations of LNP-encapsulated VEGF-A and FGF1 mRNAs combination 632 as a potential future treatment for DFU.” The last paragraph of the discussion does a very decent job in summing up the most crucial and key findings of their study while drawing the path for the other in vivo and human studies moving forward. All that being said, I believe this paragraph does not belong in the discussion and can be labeled under the “Conclusions” section.

Comments on the Quality of English Language

Overall, the whole manuscript, specifically the abstract, introduction and discussion need serious revisions and reconstructions regarding their English grammar and choices of words. I highly advise the authors take aid from online English-grammar-checking programs to significantly improve their manuscript. The authors can also take aid from the MDPI online English-grammar checking tool. 

Author Response

The response have been uploaded as an attached file. The authors of this study would like to thank you all your comments and suggestions, which have been useful to improve the quality of the publication.

Reviewer 2 Report

Comments and Suggestions for Authors

Manuscript number: cells-2840722

Title: Combination of Vascular Endothelial Growth Factor A (VEGF-A) and Fibroblast Growth Factor 1 (FGF1) Modified mRNA Improves Wound Healing in Diabetic Mice.

The author suggests that the combination of VEGF-A and FGF1 could serve as a more effective treatment for enhancing wound healing in diabetic conditions. mRNA forms of VEGF-A and FGF1 are proposed as a delivery method for treatment. The authors conducted well-designed experiments to explore the effects of the combination and elucidate the underlying mechanism. I would like to suggest that addressing some concerns may help the authors improve their manuscript.

Major:

1.     The author observed that the combination of VEGF-A and FGF1 enhances the wound healing process in the diabetic model. However, considering the proangiogenic effect of both VEGF-A and FGF1, it would be challenging to agree on the efficacy of VEGF-A and FGF1 significantly affecting DUF. To strengthen this argument, a skin wound assay on normal mice will be conducted to further support the specific impact of VEGF-A and FGF1 on DUF. The author proposed that FN1, SPP1, FGF1, and F2R were identified as up-regulated genes by VEGF-A and FGF1 treatment in Fig 5C, but they were compared to the control group. They should be compared to mono-treatment groups.

2.     The results regarding RNA-seq need to be rewritten. What is PC2 in Fig 4B? Figs 4C-E are vague. Please add a legend indicating which one is compared to control, VEGF, FGF1, and the combination. I suggest removing Table 2, as it is already stated in Figs 4 C-E. It would be beneficial to list upregulated genes depending on treatment and time points, such as those constantly increased up to day 7.

Minor:

1.     Revise English in lines 482-483.

2.     Please add full information about peptide sequences and mRNA sequences regarding VEGF-A and FGF1.

Comments on the Quality of English Language

The language of the manuscript is OK.

Author Response

Authors would like to thank the reviewer 2 for his comments and suggestions after revision, which have been very useful to improve the quality of the manuscript. The answers to your comments have been uploaded in an attached world file.

Kind regards,

Sandra.

Round 2

Reviewer 1 Report

Comments and Suggestions for Authors

In my initial review of this paper, I had asked the authors to make a few changes to their title to make it more accessible and understandable for all readers. The authors have made all the requested changes to their title.

Throughout their paper, there were some typos made by the authors that I asked them to fix and get aid from English correction tools which the authors have done so accordingly.

In the abstract of the paper, there were multiple details that were missing and some information was overfed to the readers. I had multiple suggestions to make their abstract more reader-friendly and more comprehensible without leaving any key data behind, the authors have listened to my suggestions and now, their abstract is way easier to comprehend.

I had 3 keywords in mind that I thought were missing from the list of keywords. the authors have fortunately added them to their revised manuscript.

Regarding the introduction of this paper, the structure needed serious reconstruction. The last paragraph involved all the key methodology details and results/conclusions. I suggested that the authors get rid of these inappropriately-placed details and instead add a "conclusion" section at the end of their paper. I asked the authors to use their limited word count in the introduction to showcase a clear and interesting research gap that they have found in the literature, followed by a proper investigation into similar studies to find out if their hypotheses are worth any further investigation or not. I am glad to report that the authors have significantly improved their introduction and have gotten rid of all the unnecessary methods/result data in their introduction and have moved them into their right places.

One of my biggest concerns with this paper was regarding their methodologies, when I noticed that the authors had completely forgotten to mention any ethical committee approvals for their ex vivi and in vivo investigations. Thankfully the authors have included these data in their revised manuscript.  

Another misleading feature of this paper was the absence of a proper description and explanation regarding the number of animal models in each group and a detailed timeline of all the investigations executed on them. The authors have answered all of my questions regarding their animal models and have improved the methodology section of their paper.

Regarding the surgery that they performed, I asked the authors to be more specific and go into full detail about every step of their surgical procedure. The authors have done so accordingly.

Regarding their figures and tables, figure 5 needed some quality upgrades in some of its sections which the authors have improved accordingly.

Overall, I am pretty pleased with all the improvements made by the authors and believe that their paper has significant potential. 

Reviewer 2 Report

Comments and Suggestions for Authors

Manuscript Number: cells-2840722

Title: Combination of vascular endothelial growth factor A (VEGF-A) and fibroblast growth factor 1 (FGF1) modified mRNA improves wound healing in diabetic mice.

My comment and minor errors have been addressed by Dr. Sandra Tejedoret al. in the revised manuscript.

All the best,

Kyuyeon